# Autophagic flux is impaired in the brain tissue of Tay-Sachs disease mouse model

**Tugce Sengul**[1☯], **Melike Can**[1☯], **Nurselin Ateş**[1☯], **Volkan Seyrantepe**[1,2]*

**1** İzmir Institute of Technology, Department of Molecular Biology and Genetics, İzmir, Turkey, **2** İzmir Institute of Technology, IYTEDEHAM, İzmir, Turkey

☯ These authors contributed equally to this work.
* volkanseyrantepe@iyte.edu.tr

**Data Availability Statement:** All relevant data are within the manuscript and its Supporting Information files.

**Funding:** This study is funded by TUBİTAK Grant No: 215Z083. The funders had no role in the study

## Abstract

Tay-Sachs disease is a lethal lysosomal storage disorder caused by mutations in the HexA gene encoding the α subunit of the lysosomal β-hexosaminidase enzyme (HEXA). Abnormal GM2 ganglioside accumulation causes progressive deterioration in the central nervous system in Tay-Sachs patients. *Hexa-/-* mouse model failed to display abnormal phenotype. Recently, our group generated *Hexa-/-Neu3-/-* mouse showed severe neuropathological indications similar to Tay-Sachs patients. Despite excessive GM2 ganglioside accumulation in the brain and visceral organs, the regulation of autophagy has not been clarified yet in the Tay-Sachs disease mouse model. Therefore, we investigated distinct steps of autophagic flux using markers including LC3 and p62 in four different brain regions from the *Hexa-/-Neu3-/-* mice model of Tay-Sachs disease. Our data revealed accumulated autophagosomes and autophagolysosomes indicating impairment in autophagic flux in the brain. We suggest that autophagy might be a new therapeutic target for the treatment of devastating Tay-Sachs disease.

## Introduction

Tay-Sachs disease (TSD) is one of the lysosomal storage diseases (LSD) caused by a mutation in the HexA gene, encoding for the α subunit of lysosomal β-hexosaminidase A (HEXA), responsible for the degradation of GM2 to GM3 ganglioside [1]. Consequently, in TSD pathology, accumulated GM2 in neurons leads to neuronal death and progressive neurodegeneration in affected children. Tay-Sachs patients also have developmental delay, muscle weakness, spasticity, dementia, blindness, and epilepsy followed by death at the age of two to four. *Hexa-/-* mice were generated for further thought of TSD's pathophysiology [2, 3]. However, *Hexa-/-* mice did not show any neurological phenotype although the presence of limited GM2 ganglioside accumulation in neurons. Recently, we generated a mice model with a combined deficiency of *Hexa* and *Neu3* genes which showed abnormalities in the size and numbers of lysosomes in all tissues studied, especially in the brain due to abnormal GM2 accumulation. *Hexa-/-Neu3-/-* undergo progressive neurodegeneration with neural loss and Purkinje cell depletion and survived up to 5 months. In addition, neuroinflammation has been postulated

design, data collection, and analysis, decision to publish, or preparation of the manuscript.

**Competing interests:** The authors have declared that no competing interests exist.

as a pathophysiological mechanism due to the excessive activation of glial cells and the infiltration of numerous inflammatory cells in the brain of *Hexa-/-Neu3-/-* mouse [4, 5].

In mammalian cells, two major mechanisms are carried out for the degradation of intracellular proteins: the ubiquitin-proteasome system (UPS) and autophagy [6]. In particular, autophagy is involved in lysosome-dependent pathways for damaged organelles, unfolded proteins, and accumulated cellular components to maintain cellular homeostasis [7]. Autophagic flux includes vesicle trafficking and a network in which newly produced autophagosomes (double-membrane vesicles) are fused with lysosomes to degrade autophagic cargo. In this process, autophagore formation, autophagosome completion with the closure of the membrane, and autophagosome-lysosome fusion (autolysosome) take place respectively [8]. Each step of autophagic flux is finely regulated by specific protein complexes and the initiation step, the sequestering of autophagic cargo within an isolation membrane (phagophore), is controlled by the Beclin-1 [9]. Autophagy-related (Atg) proteins generate phagophore assembly sites and enable the envelopment of cytoplasmic material. In particular, the Atg9 protein is a key regulator of autophagy induction. During the maturation of autophagosome, Atg7 has involved in the conversion of the cytosolic form of microtubule-associated protein 1 light chain 3 (LC3-I) to LC3-II which is located on both inner and outer autophagosomal membranes [9]. LC3 involves in phagophore edge folding resulting in autophagosome formation. After the formation of autolysosomes formed by the combination of autophagosomes and lysosomes, LC3-II on the outer membrane is converted back to LC3-I and then intra-autophagosomal LC3-II is degraded by lysosomal hydrolyses. Therefore, the level of LC3-II as a marker of dynamic autophagosomal membranes is generally studied to monitor autophagic activity [10]. In addition, the level of ubiquitin-binding scaffold protein p62 (aggregated endogenous substrates) which know to be associated with LC3-II in the autophagosome is mostly evaluated as a marker of termination of autophagy [9, 11]. A bunch of studies also showed a defect in autophagic flux and secondary accumulation of autophagic substrates such as autophagosomes in several LSDs [6, 12–14].

Gangliosides, known as glycosphingolipids in humans, that are mainly found in membranes of neurons contribute to promoting axon-myelin interactions, activation of trans-membrane receptor signaling, and $Ca^{2+}$ homeostasis [15–19]. However, the precise molecular mechanisms underlying their physiological or pathological activities are poorly understood. It has been shown that gangliosides released under pathological conditions may induce autophagic cell death of astrocytes in vitro [20]. In addition, *Matarrese et al.* have shown that GD3 ganglioside actively contributes to the biogenesis and maturation of autophagic vacuoles upon induction [21]. Activation of autophagy-dependent α-Syn clearance using GM1 ganglioside was also demonstrated in experimental models of Parkinson's disease (PD) *in vivo* and *in vitro* [22]. GM2 ganglioside, on the other hand, is an intermediate substrate for biosynthesis and degradation of complex brain gangliosides such as GM1a, GD1a, GD1b, and GT1b [23]. Therefore, the effect of abnormally accumulated GM2 ganglioside in neurons is important for cellular processes including autophagy flux. In the current work, we examined whether accumulated GM2 ganglioside in lysosomes causes alteration in autophagic machinery in four brain regions of early-onset TSD mice model by qRT-PCR, Western Blot, and immunohistochemical techniques. Dysfunctional autophagy was demonstrated in *Hexa-/-Neu3-/-* mice as indicated by the increase in LC3-II and accumulation of autophagosomes. Our results provide a guide for future work that elucidates the contribution of altered autophagy to TSD pathology.

## Materials & methods

### Animals

*Hexa-/-*, *Neu3-/-*, and *Hexa-/-Neu3-/-* knock-out mice models were generated as previously described [4]. Breeding and maintenance of all mice were supplied in the Turkish Council on Animal Care (TCAC) accredited animal facility of Izmir Institute of Technology according to the TCAC guidelines. Animal care was granted by the Animal Care and Use Committee of the Izmir Institute of Technology, Izmir, Turkey. Mice were housed under constant conditions (12 h light: dark cycle, room temperature 21±1˚C with water and food available *ad libitum*). Pups were weaned three weeks after birth and genotyped [4], by PCR for Hexa (F:5'-GGCCA-GATACAATCATACAG-3', PGK:5'-CACCAAAGAAGGGAGCCG GT-3', R:5'-CTGTCCA-CATACTCTCCCCACAT-3') and Neu3 (F:5'-CTCTTCTTCATTGCC GTGCT-3', NeoF:5'-GCCGAATATCATGGTGGAAA-3', R:5'-GACAAGGAGAGCCTCTGG TG-3') allele from genomic DNA. PCR protocol was followed by these conditions: 1 cycle 30 seconds at 95˚C; 30 cycles 30 seconds at 95˚C, 45 seconds at 60˚C,45 seconds at 72˚C; and 1 cycle 5 minutes at 72˚C. *WT*, *Hexa-/-*, *Neu3-/-*, *Hexa-/-Neu3-/-* mice were sacrificed at 2- and 5-month-old. The cortex, cerebellum, thalamus, and hippocampus were separated and snap-frozen with liquid nitrogen, then kept at −80˚C.

### qRT-PCR analysis

Total RNA was extracted from the cortex, cerebellum, thalamus, and hippocampus from 2- and 5-month-old *WT*, *Hexa-/-*, *Neu3-/-*, *Hexa-/-Neu3-/-* mice (n = 3) using Trizol Reagent (GeneAid) and cDNA was synthesized using reverse transcription kit (Applied Biosystems) following manufacturer's instructions. Relative mRNA expression analysis of the following autophagy-related genes, Beclin-1, Atg9, Atg7, and p62, were analyzed by Roche LightCycler 96 machine using Real-Time SYBER green PCR master mix (Roche, Swiss) with these conditions: initial denaturation at 95˚C for 10 minutes; 45 cycles at 95˚C for 20 seconds and 60˚C for 15 seconds. GAPDH gene expression was used as an endogenous control. The primers used for expression analysis are listed in Table 1.

### Western blot analysis

Protein isolation was performed from the cortex, cerebellum, thalamus, and hippocampus tissues from 2- and 5-month-old *WT*, *Hexa-/-*, *Neu3-/-*, *Hexa-/-Neu3-/-* mice (n = 3) using cold lysis buffer (50 Mm Tris- HCI, 150 mM NaCI, 1% TritonX-100, 50 mM HEPES, 10% glycerol) in the presence of protease inhibitors (Roche) for 1 hour. After the determination of protein concentration with Bradford protein assay, an equal amount of protein was loaded on 10% SDS-PAGE, followed by transfer to nitrocellulose membrane (Bio-Rad, USA). After blocking of membranes in PBS-T containing 0.1% Tween-20 and 5% dry milk for 1 hour, the membranes were incubated with anti-Beclin1 (1:2000, Santa Cruz, USA), anti-LC3 (1:1000, novus-bio, USA), anti-p62 (1:1000, Thermo, USA), and anti-Actin (1:1000, Cell Signalling, USA) primary antibodies at room temperature for 1 hour. Incubation of membranes with HRP-conjugated secondary antibodies (Jackson ImmunoResearch Lab, USA) was done at room temperature for 1 hour. The proteins were visualized by LuminataTM Forte WesternHRP Substrate (Millipore, USA) on a digital imaging system (Fusion SL, Vilber). The density of bands was normalized to actin and quantified using NIH ImageJ [4].

**Table 1. Primers that are used in RT-PCR.**

| Gene | Primer Sequences |
|---|---|
| Beclin-1 | F:5'-GAGGAGCAGTGGACAAAAGC-3' |
| | R: 5'-CAAACATCCCCTAAGGAGCA-3' |
| Atg9 | F:5'-GTGCTTATTGCCCTCACCAT-3', |
| | R: 5'-GGCATGTAGTGGATGTGTGC-3' |
| Atg7 | F: 5'-GTCGTCTTCCTATTGATGGACACC-3' |
| | R: 5'-CAAAGCAGCATTGATGACCAGC-3' |
| p62 | F: 5'-TGTGGAACATGGAGGGAAGAG-3' |
| | R: 5'-TGTGCCTGTGCTGGAACTTTC -3' |
| GAPDH | F:5'-CCCCTTCATTGACCTCAACTAC-3' |
| | R:5'-ATGCATTGCTGACAATCTTGAG-3' |

## Immunohistochemical analysis

5-month-old *Hexa-/-* and *Hexa-/-Neu3-/-* mice (n = 3) were anesthetized and perfused through the heart with 0.9% NaCl and 4% Para-formaldehyde. Brains were incubated in the fixative solution overnight at 4˚C and the following day sucrose density gradient (10%, 20%, and 30% sucrose in PBS) was applied. Embedded brains in OCT (Sigma) were sectioned coronally (10μm) using Leica Cryostat (CM1850-UV) at -20˚C. Sections were stained with anti-LC3 (8ug/ul, R&D Systems), anti-Lamp1 (1:500, Abcam), and anti-p62 (1:500, Thermo) primary antibodies overnight at 4˚C. The binding of primary antibodies was visualized using goat-anti-rat (IgH H&L AlexaFluor 568) for anti-LC3 and anti-rabbit (IgG Alexa Fluor 488) for anti-Lamp1 and anti-p62. The slides were mounted with Fluoroshield mounting medium with DAPI (Abcam, USA) The images were obtained by Fluorescence Microscopy (Olympus-BX53F). Co-localization analysis of red and green fluorescence was measured using ImageJ.

## Statistical analysis

GraphPad Prism 7 (v. 7.0a, GraphPad Software, Inc) was used to perform all statistical analyses. *WT*, *Hexa-/-*, *Neu3-/-*, and *Hexa-/-Neu3-/-* groups were compared by one-way ANOVA. All values are presented as mean ± SEM.

## Results

### Age- and brain region-specific alterations in the initiation of autophagic flux

To assess whether there is an alteration in the initiation of autophagic flux, the expression levels of autophagore formation-related genes, Beclin-1 and Atg9, were studied in the cortex, cerebellum, thalamus, and hippocampus of 2- and 5-month-old *WT*, *Hexa-/-*, *Neu3-/-* and *Hexa-/-Neu3-/-* mice by qRT-PCR. Although the expression level of Beclin-1 in 2-month-old *Hexa-/-Neu3-/-* mice displayed a slight decrease in both cortex (Fig 1A) and hippocampus (Fig 1M) compared to *WT* mice, we found a significantly increased level in the cerebellum (Fig 1E) and thalamus (Fig 1I). Additionally, no significant difference in the expression level of Beclin-1 was detected in 3-month-old *WT*, *Hexa-/-*, *Neu3-/-* and *Hexa-/-Neu3-/-* (S1 Fig). A higher expression level of another autophagore formation-related gene, Atg9, is detected in the cerebellum of 2-month-old *Neu3-/-* and *Hexa-/-Neu3-/-* mice compared to *WT* mice (Fig 1G), but no obvious quantitative differences were observed in the cortex (Fig 1C), thalamus (Fig 1K) and hippocampus (Fig 1O). We also showed significantly elevated expression levels of Beclin-1

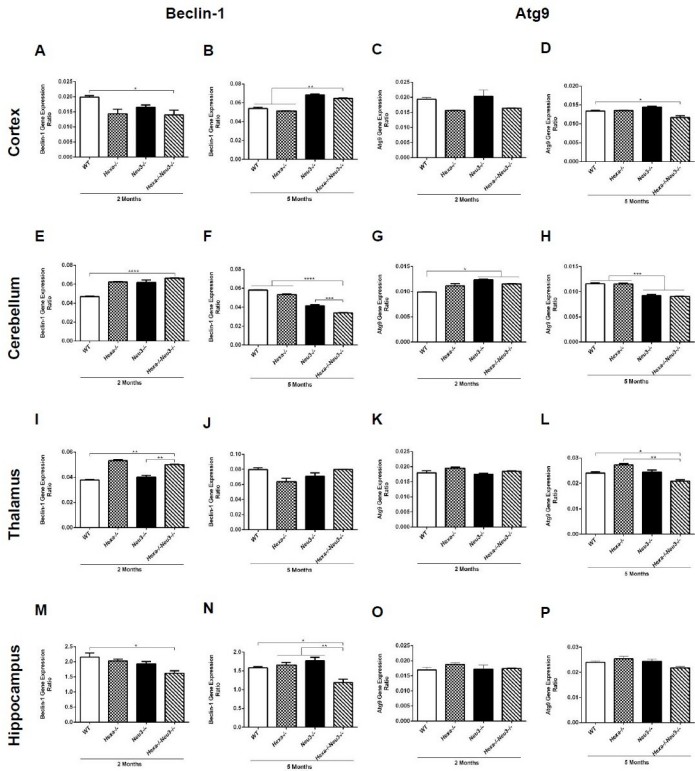

**Fig 1.** Beclin1 and Atg9 gene expression levels in cortex (A-D), cerebellum (E-H), thalamus (I-L), and hippocampus (M-P) of 2- and 5-month-old *WT*, *Hexa-/-*, *Neu3-/-*, *Hexa-/-Neu3-/-* mice. Expression ratios were calculated by the ΔCT method and percent ratios were represented. One-way ANOVA analysis was used to determine p-values via GraphPad. Data are reported as mean ± SEM (n = 3; *p<0,05, **p<0,025, ***p<0,01, ****p<0,001).

in the cortex of 5-month-old *Neu3-/-* and *Hexa-/-Neu3-/-* mice compared to *WT* and *Hexa-/-* mice (Fig 1B). Interestingly, *Hexa-/-Neu3-/-* mice displayed lower expression levels of Beclin-1 compared to age-matched mice groups in both cerebellum (Fig 1F) and hippocampus (Fig 1N). In addition, decreased expression level of Atg9 was observed in the cortex (Fig 1D), cerebellum (Fig 1H), and thalamus (Fig 1L) of 5-month-old *Hexa-/-Neu3-/-* mice, but not in the hippocampus (Fig 1P). The decreased expression level of Beclin-1 and Atg9 in older age *Hexa-/-Neu3-/-* mice indicates an alteration in the initiation of autophagic flux.

We also performed western blot analysis to detect the protein level of Beclin-1 in the different brain regions of 2-and 5-month-old mice. Significantly decreased levels of Beclin-1 were detected in the cerebellum (Fig 2F) and hippocampus (Fig 2L) but not in the cortex (Fig 2C) and thalamus (Fig 2I) of 5-month-old *Hexa-/-Neu3-/-* mice compared to *WT* and *Hexa-/-*. No significant difference in the protein level of Beclin-1 was detected in 2 and 3-month-old *WT*, *Hexa-/-*, *Neu3-/-* and *Hexa-/-Neu3-/-* (Figs 2B, 2E, 2H, 2K and S2). Consisted with qRT-PCR analysis, Western Blot analysis showed alterations in the initiation of the autophagic flux of *Hexa-/-Neu3-/-* mice.

## Increased autophagosome number in the elongation of autophagic flux

Significantly increased expression levels of the Atg7 gene were determined in the cortex (Fig 3A), cerebellum (Fig 3C), and thalamus (Fig 3E) of 2-month-old *Hexa-/-Neu3-/-* compared to *Hexa-/-* mice. In addition, we showed that there is an increased expression level of Atg7 in the

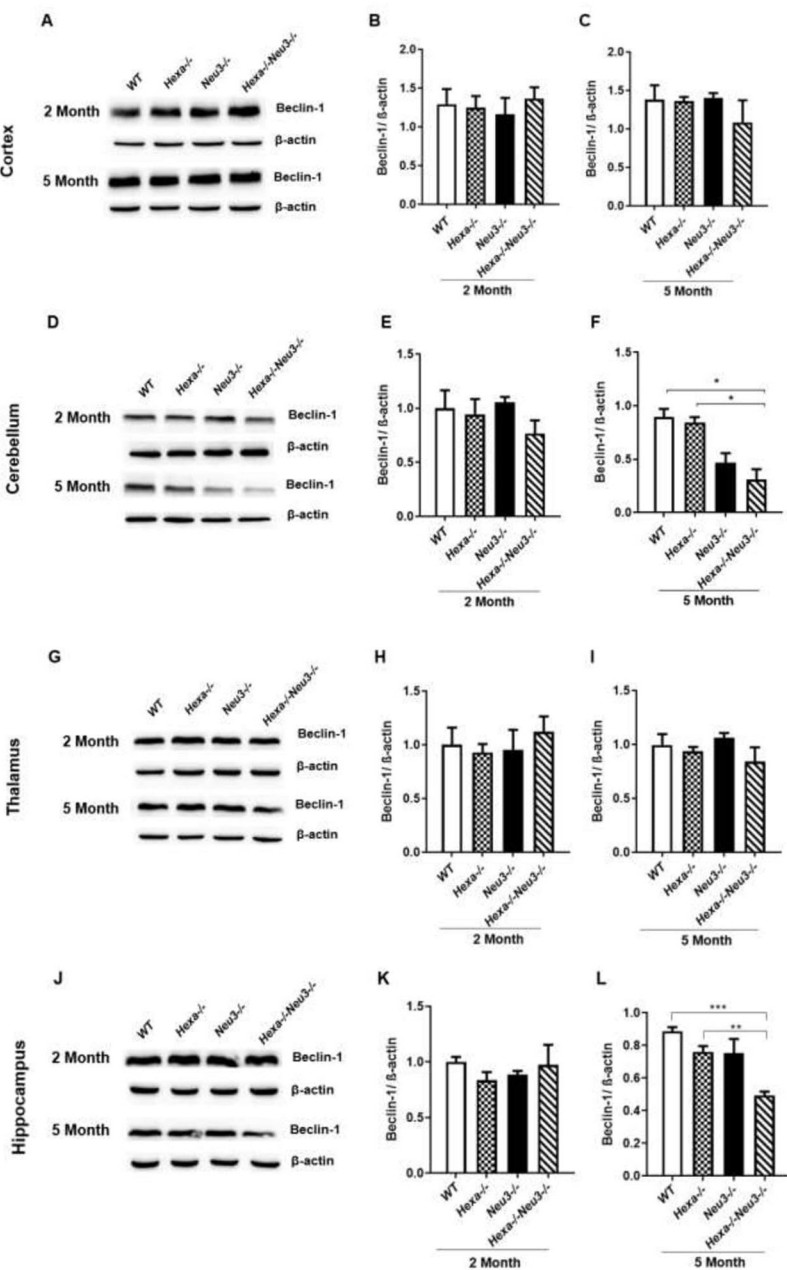

**Fig 2.** Immunoblot images and densitometric analysis of anti-Beclin-1 in the cortex (A-C), cerebellum (D-F), thalamus (G-I), and hippocampus (J-L) of 2- and 5-month-old *WT*, *Hexa-/-*, *Neu3-/-*, *Hexa-/-Neu3-/-* mice. β-actin was an internal control. Band intensities were determined by ImageJ and p values were determined by One-way-ANOVA analysis by GraphPad. Data are reported as mean ± SEM (n = 3; *p<0,05, **p<0,025, ***p<0,01).

cerebellum (Fig 3D) and hippocampus (Fig 3H) but not in the cortex (Fig 3B) and thalamus (Fig 3F) of 5-month-old *Hexa-/-Neu3-/-* mice compared to *WT*. Compared to *WT* and *Hexa-/-* mice, we showed significantly elevated protein levels of autophagosome-related marker LC3 in all brain regions studied in 5-month-old *Hexa-/-Neu3-/-* mice (Figs 4C, 4F, 4I, and 4L). Although we found notably increased LC3 protein levels in the cortex (Fig 4B) and cerebellum (Fig 4E) regions of 2-month-old *Hexa-/-Neu3-/-* mice compared to *WT* and

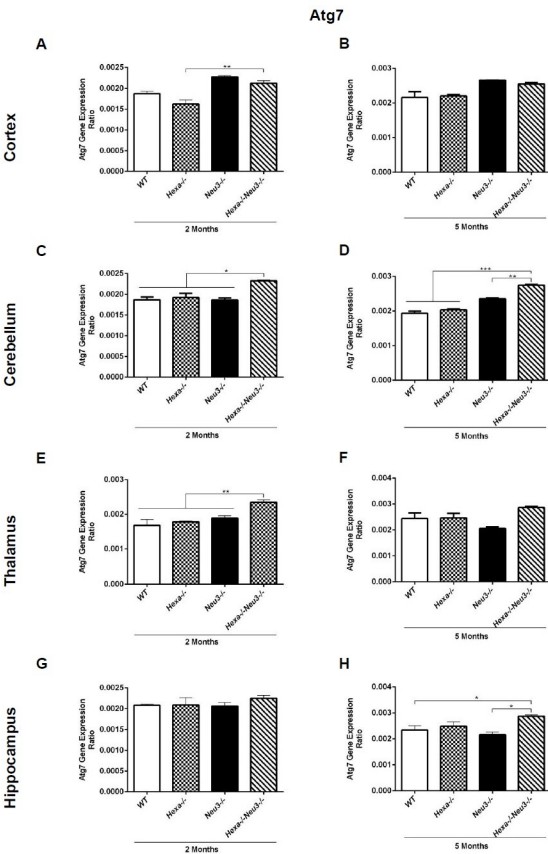

**Fig 3.** Atg7 gene expression of the levels of the cortex (A, B), cerebellum (C, D), thalamus (E, F), and hippocampus (G, H) of 2- and 5-month-old *WT*, *Hexa-/-*, *Neu3-/-*, *Hexa-/-Neu3-/-* mice. Expression ratios were calculated by the ΔCT method and percent ratios were represented. One-way ANOVA analysis was used to determine p-values via GraphPad. Data are reported as mean ± SEM (n = 3; *p<0,05, **p<0,025, ***p<0,01).

*Hexa-/-* mice, there was no significant difference in the thalamus (Fig 4H) and hippocampus (Fig 4K). Using immunohistochemical analysis we confirmed an increased number of LC3(+) vesicles in all brain regions of 5-month-old *Hexa-/-Neu3-/-* mice compared to *Hexa-/-* mice (Fig 5E–5H) and we observed no difference in 2-month-old *WT*, *Hexa-/-*, *Neu3-/-* and *Hexa-/-Neu3-/-* (S3 Fig). In addition, we detected a significant elevation in LC3(+) vesicles colocalized with LAMP1 in the cortex, cerebellum, and hippocampus but not in the thalamus of 5-months-old *Hexa-/-Neu3-/-* mice compared to *Hexa-/- mice* (Fig 5I–5L).

## Significantly increased levels of p62/SQSTM1 in termination of autophagic flux

To evaluate the termination of autophagic flux, the expression level of p62 was analyzed by qRT-PCR. *Hexa-/-Neu3-/-* mice displayed significantly higher p62 expression compared to *WT* and *Hexa-/-* counterparts in the cerebellum (Fig 6C and 6D), thalamus (Fig 6E and 6F), and hippocampus (Fig 6G and 6H) in both age groups but no obvious differences were detected in cortex region (Fig 6A and 6B). In parallel to our qRT-PCR results, we observed a significantly increased level of p62 in the cerebellum (Fig 7F), thalamus (Fig 7I), and hippocampus (Fig 7L) of 5-month-old *Hexa-/-Neu3-/-* mice compared to age-matched *WT* and *Hexa-/-* mice using western blot. However significant changes were not observed in the cortex

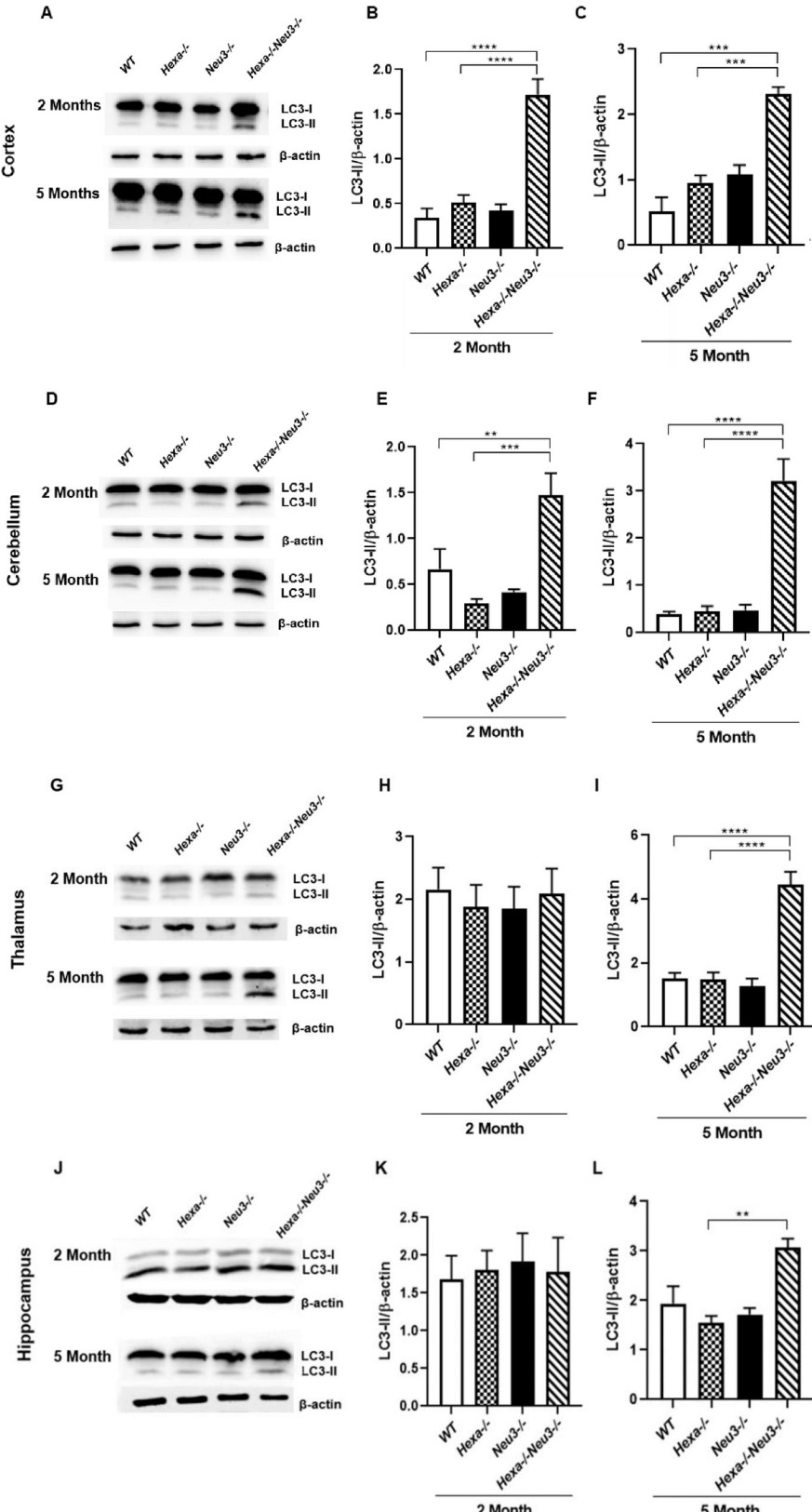

**Fig 4.** Immunoblot images and densitometric analysis of the nti-LC3 in the cortex (A-C), cerebellum (D-F), thalamus (G-I), and hippocampus (J-L) of 2- and 5-month-old *WT*, *Hexa-/-*, *Neu3-/-*, *Hexa-/-Neu3-/-* mice. β-actin was an

internal control. Band intensities were determined by ImageJ and p values were determined by One-way-ANOVA analysis by GraphPad. Data are reported as mean ± SEM (n = 3; *p<0,05, ***p<0,01, ****p<0,001).

among 5-month-old mice (Fig 7C). A significantly elevated level of p62 was shown in both cortex (Fig 7B) and hippocampus (Fig 7K) of 2-month-old *Hexa-/-Neu3-/-* mice compared to both *WT* and *Hexa-/-*. In addition, the p62 protein level was high in the cerebellum (Fig 7E) and thalamus (Fig 7H) of 2-month-old *Hexa-/-Neu3-/-* mice compared to age-matching *Hexa-/-* mice. We found that the immunohistochemical analysis was consistent with Western blot data confirming a significantly increased level of p62, especially in the cerebellum and hippocampus of 5-month-old *Hexa-/-Neu3-/-* mice compared to *Hexa-/-* (Fig 8B and 8D, 8H). p62 aggregation was slightly but not significantly higher in the thalamus (Fig 7E) of 5-month-old *Hexa-/-Neu3-/-* mice compared to *Hexa-/-* mice. On the other hand,

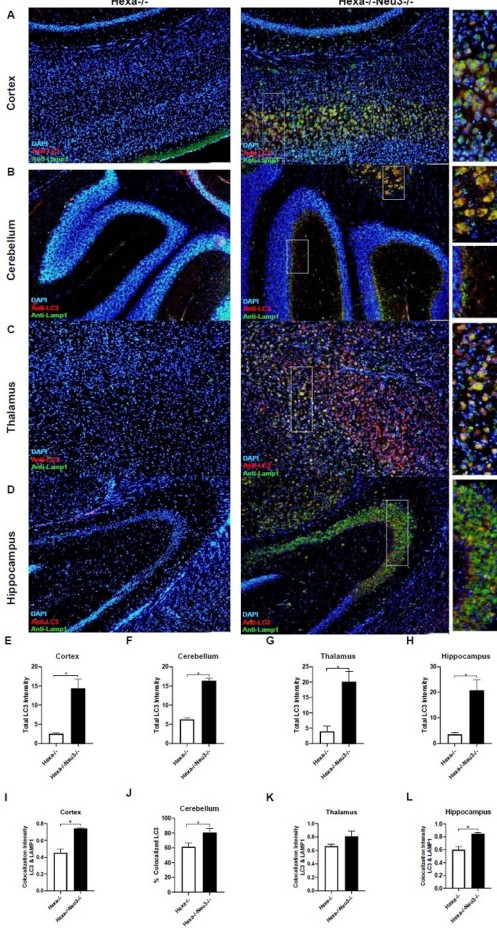

**Fig 5.** Immunohistochemical colocalization analysis images for cortex (A), cerebellum (B), thalamus (C), and hippocampus (D) sections from 5 months old *Hexa-/-* and *Hexa-/-Neu3-/-* mice. The sections were stained with anti-LC3 antibody (red; Autophagosome marker), anti-Lamp1 (green; lysosomal marker), and DAPI (blue; nucleus). Total LC3 intensity was represented for cortex (E), cerebellum (F), thalamus (G), and hippocampus (H). A yellow signal signifies the colocalization of LC3 and Lamp1 as autophagolysosomes. Colocalization percentages LC3 to Lamp1 were represented for cortex (I), cerebellum (J), thalamus (K), and hippocampus (L) Scale bar = 20 μm. The data are represented as the mean ± S.E.M. Unpaired t-test was used for statistical analysis. (*p<0.05, **p<0.025, ***p<0.01 and ****p<0.001).

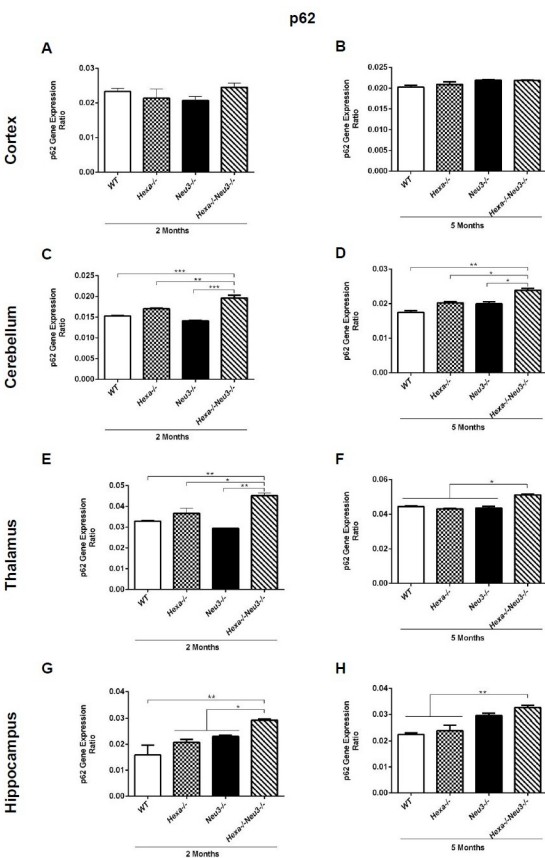

**Fig 6.** p62 gene expression levels of the cortex (A, B), cerebellum (C, D), thalamus (E, F), and hippocampus (G, H) of 2- and 5-month-old *WT*, *Hexa-/-*, *Neu3-/-*, *Hexa-/-Neu3-/-* mice. Expression was calculated by the ΔCT method and percent ratios were represented. One-way ANOVA analysis was used to determine p-values via GraphPad. Data are reported as mean ± SEM (n = 3; $^*$p<0,05, $^{**}$p<0,025, $^{***}$p<0,01).

immunohistochemical analysis did not show p62 accumulation in brain of 2-month-old *WT*, *Hexa-/-*, *Neu3-/-* and *Hexa-/-Neu3-/-* (S4 Fig).

## Discussion

Lysosomal dysfunction resulting from the accumulation of endogenous substrates in lysosomes is the hallmark of LSDs including GM2 gangliosidosis [24]. A bunch of studies on LSDs reported not only lysosomal storage but also impairment of other cellular processes such as calcium homeostasis, lipid synthesis, and signaling pathways [24–26]. Owing to the essential role of lysosomes in autophagy, it was reasonable to expect that accumulated GM2 ganglioside in lysosomes of *Hexa-/-Neu3-/-* mice could have an effect on autophagic flux. However, the relation between accumulated GM2 ganglioside and the autophagy process has not been identified. In the present study, we investigated whether autophagic flux is altered in the brain of *Hexa-/-Neu3-/-* mice correlated with insufficient lysosomal due to accumulation of GM2 ganglioside.

Beclin-1 activation is a marker of autophagy induction which triggers autophagore formation [27]. Although the elevated level of Beclin-1 has been shown in *npc1-/-* mice, fibroblasts of Niemann-Pick patients [27], and *β-gal-/-* mouse brain of GM1 gangliosidosis [12], the level of Beclin-1 has not changed in mice model of multiple sulphatase deficiency (MSD) and

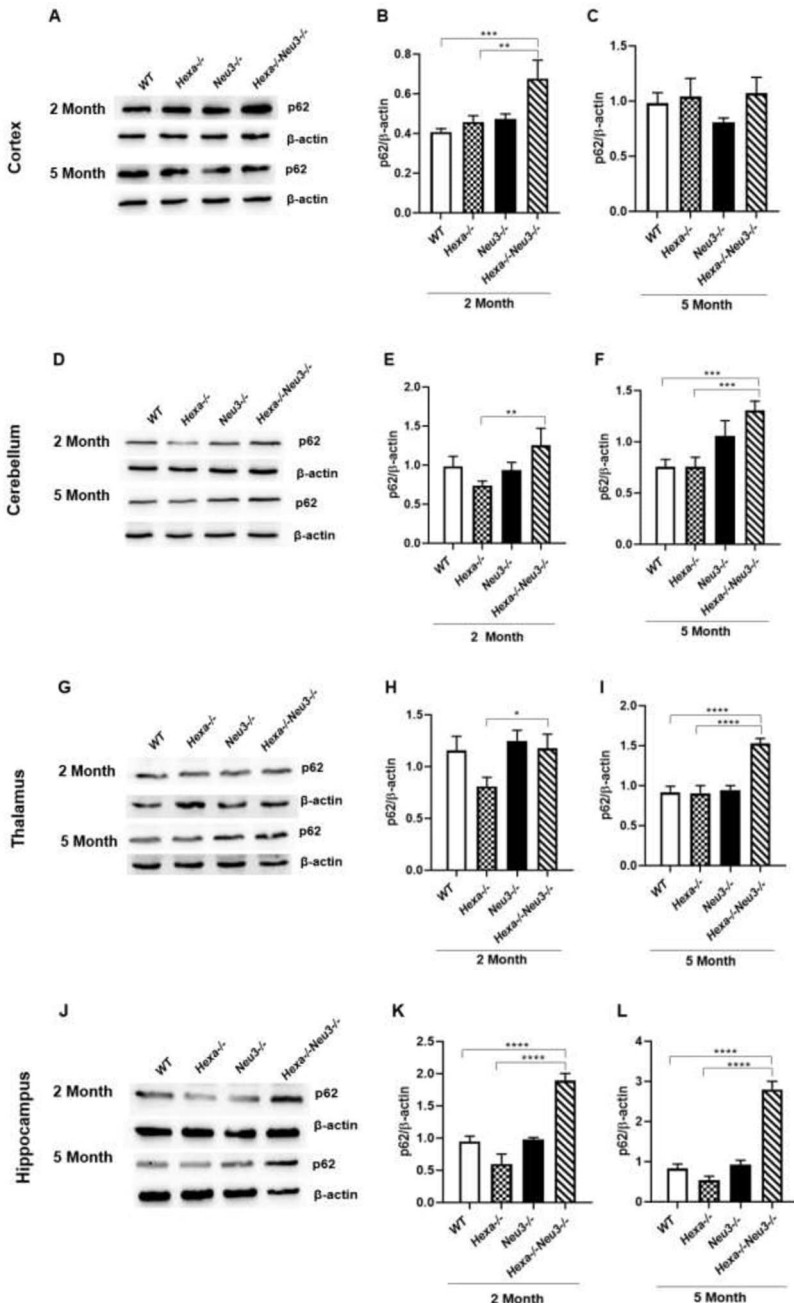

**Fig 7.** Immunoblot images and densitometric analysis of anti-p62 in the cortex (A-C), cerebellum (D-F), thalamus (G-I), and hippocampus (J-L) of 2- and 5-month-old *WT*, *Hexa-/-*, *Neu3-/-*, *Hexa-/-Neu3-/-* mice. β-actin was an internal control. Band intensities were determined by ImageJ and p values were determined by One-way-ANOVA analysis by GraphPad. Data are reported as mean ± SEM (n = 3; *p<0,05, **p<0,025, ***p<0,01, ****p<0,001).

mucopolysaccharidosis type IIIA (MPS-IIIA) [6]. Here, we demonstrated that both mRNA and protein levels of Beclin-1 decreased in the hippocampus and cerebellum of 5-month-old *Hexa-/-Neu3-/-* mice (Figs 1B, 2F and 2L). In parallel, the lower expression of Atg9 was detected in the cortex (Fig 1D), cerebellum (Fig 1H), and thalamus (Fig 1L) of 5-month-old *Hexa-/-Neu3-/-* mice. These results indicate an alteration in autophagic pathway initiation.

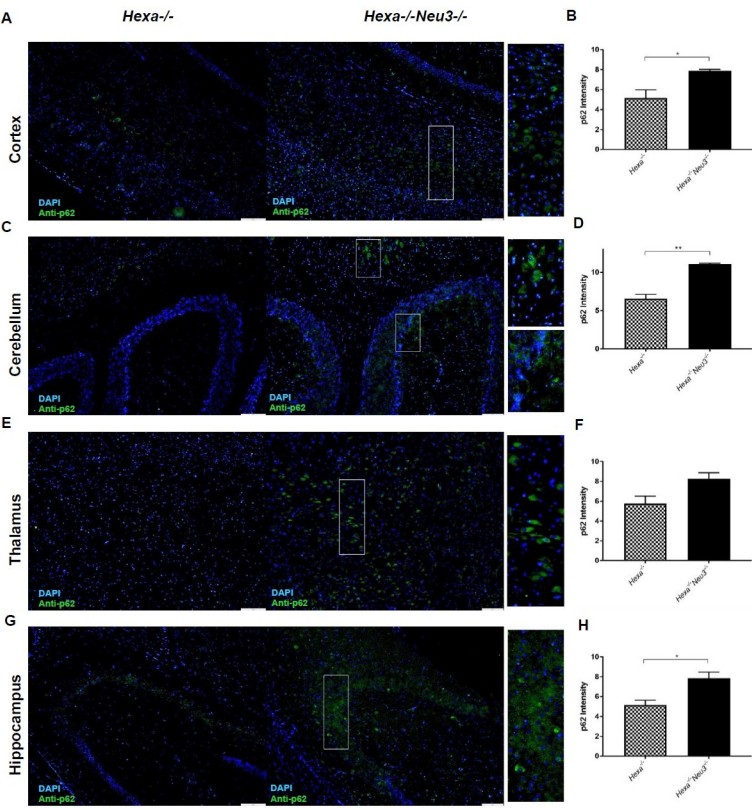

**Fig 8.** Immunohistochemical analysis images for cortex (A), cerebellum (C), thalamus (E), and hippocampus (G) sections from 5 months old *Hexa-/-* and *Hexa-/-Neu3-/-* mice. The sections were stained with anti-p62 antibody (green; Autophagic termination marker) and DAPI (blue; nucleus). Quantification of the relative increase in p62 intensity was represented for cortex (B), cerebellum (D), thalamus (F), and hippocampus (H) Scale bar = 20 μm. The data are represented as the mean ± S.E.M. One-way ANOVA was used for statistical analysis. (*$p < 0.05$, **$p < 0.025$, ***$p < 0.01$ and ****$p < 0.001$).

Autophagosome accumulation has also been reported in several types of LSDs including Danon disease [28], Pompe disease [13], mucolipidosis [14], and mucopolysaccharidoses type IIIA [6]. Increased autophagosome formation or decreased autophagosome-lysosome fusion could lead to autophagosome accumulation [6]. Consistent with previously reported studies, we demonstrated an elevated level of LC3-II in all brain regions of 5-month-old *Hexa-/-Neu3-/-* mice (Figs 4C and 4F, 4I, 4L and 5). These data suggest that *Hexa-/-Neu3-/-* mice have secondarily accumulated autophagosomes besides GM2 ganglioside accumulation. Furthermore, immunocytochemical analysis of LC3-Lamp1 colocalization also revealed elevated autophagolysosomes in *Hexa-/-Neu3-/-* mice (Fig 5). However, it is not clear whether the increase in LC3-Lamp1 colocalization is due to accumulated lysosomes or increased autophagolysosomes in *Hexa-/-Neu3-/-* mice. To clarify the mechanism of autophagosome-lysosome fusion, further investigations are required.

Previously it was shown that the level of autophagy substrates, such as p62/SQSTM1 are significantly increased in several LDS indicating an impairment of the autophagic flux. p62/SQSTM1 protein, a component of ubiquitinated protein aggregates, is responsible for targeting polyubiquitinated proteins to autophagosomes [29] and degraded in the termination of the autophagic pathway along with autophagic cargo [30]. Accumulation of p62/SQSTM1 has been reported in the endosomal/lysosomal fraction of *npc1-/-* mouse brain lysates [31],

cultured fibroblasts of Fabry patients [32], mucolipidosis type II, III [33] and type IV (MLIV) fibroblasts [34], muscle fibers of Pompe disease [35], and brain of Gaucher disease mice model [36]. Similarly, in our study, we showed accumulated p62/SQSTM1 in the brain of 5-month-old *Hexa-/-Neu3-/-* mice suggesting the impairment in the termination step of autophagic flux.

Our knowledge of molecular and cellular mechanisms for Tay-Sachs disease is mostly limited to what we have learned from skin fibroblast and iPSCs obtained from patients with Tay-Sachs disease and recently generated mice model with combined deficiency of β-hexosaminidase A and neuraminidase 3. To our knowledge, this is the first study demonstrating progressive alterations in the autophagic flux in the brain tissue of mice with Tay-Sachs disease. Our findings provide insights into the dysregulation of autophagy in the brain and suggest a potential therapeutic approach to reduce lysosomal accumulation by targeting the regulation and activation of proper autophagy. However, further *in vitro* and *in vivo* studies are necessary to elucidate the precise molecular mechanisms underlying dysregulated autophagy in neurons and glial cells of the Tay-Sachs mice model.

## Supporting information

**S1 Fig.** Beclin-1 gene expression levels of the cortex (A), cerebellum (B), thalamus (C), and hippocampus (D) of 3-month-old WT, Hexa-/-, Neu3-/-, Hexa-/-Neu3-/- mice. Expression ratios were calculated by the ΔCT method and percent ratios were represented. One-way ANOVA analysis was used to determine p-values via GraphPad. Data are reported as mean ± SEM (n = 2)
(TIF)

**S2 Fig.** Immunoblot images and densitometric analysis of anti-Beclin-1 in the cortex (A, B), cerebellum (C, D), thalamus (E, F), and hippocampus (G, H) of 3-month-old WT, Hexa-/-, Neu3-/-, Hexa-/-Neu3-/- mice. β-actin as an internal control. Band intensities were determined by ImageJ and p values were determined by One-way-ANOVA analysis by GraphPad. Data are reported as mean ± SEM (n = 2)
(TIF)

**S3 Fig.** Immunohistochemical colocalization analysis images for cortex (A), cerebellum (B), thalamus (C), and hippocampus (D) sections from 2-months-old Hexa-/- and Hexa-/-Neu3-/- mice. The sections were stained with anti-LC3 antibody (red; Autophagosome marker), anti-Lamp1 (green; lysosomal marker), and DAPI (blue; nucleus). A yellow signal signifies the colocalization of LC3 and Lamp1 as autophagolysosome. Scale bar = 20 μm
(TIF)

**S4 Fig.** Immunohistochemical analysis images for cortex (A), cerebellum (B), thalamus (C), and hippocampus (D) sections from 2-months-old Hexa-/- and Hexa-/-Neu3-/- mice. The sections were stained with anti-p62 antibody (green; Autophagic termination marker) and DAPI (blue; nucleus). Scale bar = 20 μm
(ZIP)

**S1 Raw data.**
(ZIP)

## Author Contributions

**Conceptualization:** Volkan Seyrantepe.

**Investigation:** Tugce Sengul, Melike Can, Nurselin Ateş.

**Supervision:** Volkan Seyrantepe.

**Writing – original draft:** Volkan Seyrantepe.

**Writing – review & editing:** Volkan Seyrantepe.

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
