## [Decision Letter · Decision Letter 0]

20 Jun 2022

PONE-D-22-09806AUTOPHAGIC FLUX IS IMPAIRED IN BRAIN TISSUE OF TAY-SACHS DISEASE MOUSE MODELPLOS ONE

Dear Dr. Seyrantepe,

Thank you for submitting your manuscript to PLOS ONE. After careful consideration, we feel that it has merit but does not fully meet PLOS ONE’s publication criteria as it currently stands. Therefore, we invite you to submit a revised version of the manuscript that addresses the points raised during the review process.

 In some of the LC3 blots (e.g. Fig. 4J) there are three clearly visible bands. The authors should explain which of the two lower bands was considered as LC3-II. If possible, a higher quality LC3 blots should be presented in Fig. 4J. 

Reviewer 1:

This study aims to investigate alterations in autophagic flux in the new mouse model of Tay-Sachs disease, which should provide information for future mechanistic and therapeutic studies in this field. 

1. However, this study only includes 3 mice each groups, lacking statistical rigor.

2. The implications of alterations of those gene expression are not well-explained. The rationale why the authors are looking at a certain gene is not fully explained. It is difficult to follow why change of one gene in a certain brain region means something for a better understanding of pathogenesis mechanisms of this mouse model. Moreover, how the information collected in this study will contribute to a better understanding of Tay-Sachs disease and this mouse model is not clear. Basically, changes of several genes in some brain regions (but not others) at a certain time is only an indicator (weak evidence) of something. This study lacks a clear logic, strong evidence, and a meaningful conclusion. 

3. The reviewer is not convinced that targeting autophagic flux could be potentially therapeutic, which seems to be a conclusion of this study.

4. There are many grammar mistakes.

Reviewer 2:

The authors examined autophagic flux initiation, autophagosome number in the elongation and p62/SQSTM1 in termination in 2 moth and 5 month TSD mice model cortex, cerebellum, thalamus and hippocampus by qRT-PCR, Western Blot and imminohistochemical techniques. The data revealed accumulated autophagosomes, indicating impairment in autophagic flux in the brain. This work elucidates the contribution of altered autophagy to TSD pathology. I would like to see the following points addressed in the manuscript before its publication.

1. Hexa-/-Neu3-/- undergo progressive neurodegeneration and survived up to 5 months. However, no significant difference in the protein level of Beclin-1 was detected in 2-month-old WT, Hexa-/-, Neu3-/- and Hexa-/-Neu3-/- mice. Thus, more time point data is necessary to determine when autophagic flux alteration in those gene KO mice.

2. LC3 protein level in cortex and cerebellum regions of 2-month-old Hexa-/-Neu3-/- mice was notably elevated compared to WT. It is necessary to analyse the number of LC3(+) vesicles in cortex and cerebellum using immunohistochemistry.

3. P62 protein level was significantly elevated shown in both cortex and hippocampus of 2-month-old Hexa-/-Neu3-/- mice compared to WT. It is necessary to analyse the P62 in cortex and hippocampus using immunohistochemistry.

We look forward to receiving your revised manuscript.

Kind regards,

Vladimir Trajkovic

Academic Editor

PLOS ONE

Journal Requirements:

“This study is funded by TUBİTAK Grant No: 215Z083”

Reviewers' comments:

Reviewer's Responses to Questions

**Comments to the Author**

1. Is the manuscript technically sound, and do the data support the conclusions?

Reviewer #1: Partly

Reviewer #2: Yes

2. Has the statistical analysis been performed appropriately and rigorously? 

Reviewer #1: I Don't Know

Reviewer #2: Yes

3. Have the authors made all data underlying the findings in their manuscript fully available?

Reviewer #1: Yes

Reviewer #2: No

4. Is the manuscript presented in an intelligible fashion and written in standard English?

Reviewer #1: No

Reviewer #2: Yes

5. Review Comments to the Author

Reviewer #1: This study aims to investigate alterations in autophagic flux in the new mouse model of Tay-Sachs disease, which should provide information for future mechanistic and therapeutic studies in this field.

1. However, this study only includes 3 mice each groups, lacking statistical rigor.

2. The implications of alterations of those gene expression are not well-explained. The rationale why the authors are looking at a certain gene is not fully explained. It is difficult to follow why change of one gene in a certain brain region means something for a better understanding of pathogenesis mechanisms of this mouse model. Moreover, how the information collected in this study will contribute to a better understanding of Tay-Sachs disease and this mouse model is not clear. Basically, changes of several genes in some brain regions (but not others) at a certain time is only an indicator (weak evidence) of something. This study lacks a clear logic, strong evidence, and a meaningful conclusion.

3. The reviewer is not convinced that targeting autophagic flux could be potentially therapeutic, which seems to be a conclusion of this study.

4. There are many grammar mistakes.

Reviewer #2: The authors examined autophagic flux initiation, autophagosome number in the elongation and p62/SQSTM1 in termination in 2 moth and 5 month TSD mice model cortex, cerebellum, thalamus and hippocampus by qRT-PCR, Western Blot and imminohistochemical techniques. The data revealed accumulated autophagosomes, indicating impairment in autophagic flux in the brain. This work elucidates the contribution of altered autophagy to TSD pathology. I would like to see the following points addressed in the manuscript before its publication.

1. Hexa-/-Neu3-/- undergo progressive neurodegeneration and survived up to 5 months. However, No significant difference in the protein level of Beclin-1 was detected in 2-month-old WT, Hexa-/-, Neu3-/- and Hexa-/-Neu3-/- mice. Thus, more time point data is necessary to determine when autophagic flux alteration in those gene KO mice.

2. LC3 protein level in cortex and cerebellum regions of 2-month-old Hexa-/-Neu3-/- mice was notably elevated compared to WT. It is necessary to analysis the number of LC3(+) vesicles in cortex and cerebellum using immunohistochemical.

3. P62 protein level was significantly elevated shown in both cortex and hippocampus of 2-month-old Hexa-/-Neu3-/- mice compared to WT. It is necessary to analysis the P62 in cortex and hippocampus using immunohistochemical.

6. PLOS authors have the option to publish the peer review history of their article (what does this mean?). If published, this will include your full peer review and any attached files.

Reviewer #1: No

Reviewer #2: No

---

## [Author Response · Author response to Decision Letter 0]

19 Sep 2022

Response to reviewers 

Reviewer 1:

This study aims to investigate alterations in autophagic flux in the new mouse model of Tay-Sachs disease, which should provide information for future mechanistic and therapeutic studies in this field. 

1. However, this study only includes 3 mice each groups, lacking statistical rigor.

All data presented here were replicated in at least three mice with all histochemical staining data quantified by experimenters who were blind to genotypes/treatments. Animals were randomly allocated into control and experimental groups.

2. The implications of alterations of those gene expression are not well-explained. 

The rationale why the authors are looking at a certain gene is not fully explained.

The rationale for why looking at a certain gene expression is explained in the introduction section of the manuscript as below

In mammalian cells, two major mechanisms are carried out for the degradation of intracellular proteins which are the ubiquitin-proteasome system (UPS) and autophagy [6]. In particular, autophagy is involved in lysosome-dependent pathways for damaged organelles, unfolded proteins, and accumulated cellular components to maintain cellular homeostasis [7]. Autophagic flux includes vesicle trafficking and a network in which newly produced autophagosomes (double-membrane vesicles) are fused with lysosomes to degrade autophagic cargo. In this process, autophagore formation, autophagosome completion with the closure of membrane, and autophagosome-lysosome fusion (autolysosome) take place respectively [8]. Each step of autophagic flux is finely regulated by specific protein complexes and the initiation step, the sequestering of autophagic cargo within an isolation membrane (phagophore), is controlled by the Beclin-1. Beclin-1 consists of three distinct structural domains interacting with proteins that regulate autophagy machinery. In particular, Beclin-1, which can interact with several cofactors, promotes the formation of the Beclin-1-Vps34-Vps15 core body, in this way induces the initiation of autophagic flux [9]. Autophagy-related (Atg) proteins generate phagophore assembly sites and enable the envelopment of cytoplasmic material. In particular, the Atg9 protein is a key regulator of autophagy induction. During the maturation of autophagosome, Atg7 has involved the conversion of the cytosolic form of microtubule-associated protein 1 light chain 3 (LC3-I) to LC3-II which is located on both inner and outer autophagosomal membranes [9]. LC3 is involved in phagophore edge folding that resulted in autophagosome formation. After the formation of autolysosomes formed by the combination of autophagosomes and lysosomes, LC3-II on the outer membrane is converted back to LC3-I and then intra-autophagosomal LC3-II is degraded by lysosomal hydrolyses. Therefore, the level of LC3-II as a marker of dynamic autophagosomal membranes is generally studied to monitor autophagic activity [10]. In addition, the level of ubiquitin-binding scaffold protein p62 (aggregated endogenous substrates) which know to be associated with LC3-II in the autophagosome is mostly evaluated as a marker of termination of autophagy [9,11]. A bunch of studies also showed a defect in autophagic flux and secondary accumulation of autophagic substrates such as autophagosomes in several LSDs [6,12–14].

 It is difficult to follow why change of one gene in a certain brain region means something for a better understanding of pathogenesis mechanisms of this mouse model. 

Moreover, how the information collected in this study will contribute to a better understanding of Tay-Sachs disease and this mouse model is not clear.

Basically, changes of several genes in some brain regions (but not others) at a certain time is only an indicator (weak evidence) of something. 

This study lacks a clear logic, strong evidence, and a meaningful conclusion. 

In this study, we show that abnormal GM2 accumulation in the different region of brain tissue from TSD mouse model leads to impaired autophagy evident from attenuated expression of some components of the autophagy network, decreased autophagosome formation, and reduced autophagy flux. We for the first time study the expression of some fundamental autophagic markers (LC3, p62, and Beclin-1) in a TSD murine model by q-RT-PCR, immunohistochemistry and Western blot.

3. The reviewer is not convinced that targeting autophagic flux could be potentially therapeutic, which seems to be a conclusion of this study.

We are sorry for not convincing the reviewer that targeting autophagic flux could be potentially therapeutic in this manuscript. However, we have some preliminary in vitro data indicating lithium as a therapeutic agent could be used to restore dysregulated autophagy in Tay-Sachs cells. That data is not completed yet to add to the manuscript. 

Although additional studies are required to fully characterize the autophagy machinery in the recently generated TSD murine model, this paper suggests impaired autophagy as a new aspect involved in the molecular pathogenesis of TSD. Despite important advances made in the last years in TSD research, the principally studied therapies such as gene and anti-inflammatory therapy and their combination can increase the life expectancy of the treated TSD mice (unpublished data yet) autophagy modulation could be a promising option to be tested in combination therapies to help to achieve a complete TSD phenotype rescue.

4. There are many grammar mistakes.

In this final version of the manuscript, we used a free online writing assistant called Grammarly 

Reviewer 2:

The authors examined autophagic flux initiation, autophagosome number in the elongation, and p62/SQSTM1 in termination in 2-month and 5-month TSD mice model cortex, cerebellum, thalamus and hippocampus by qRT-PCR, Western Blot, and immunohistochemical techniques. 

The data revealed accumulated autophagosomes, indicating impairment in autophagic flux in the brain. 

This work elucidates the contribution of altered autophagy to TSD pathology. I would like to see the following points addressed in the manuscript before its publication.

1. Hexa-/-Neu3-/- undergo progressive neurodegeneration and survived up to 5 months. However, no significant difference in the protein level of Beclin-1 was detected in 2-month-old WT, Hexa-/-, Neu3-/-, and Hexa-/-Neu3-/- mice. Thus, more time point data is necessary to determine when autophagic flux alteration in those gene KO mice.

We added one more time point (3-months old mice) to determine when autophagic flux alterations (Fig Supp 1 and Fig Supp 2) 

2. LC3 protein level in cortex and cerebellum regions of 2-month-old Hexa-/-Neu3-/- mice was notably elevated compared to WT. It is necessary to analyze the number of LC3(+) vesicles in cortex and cerebellum using immunohistochemistry.

We added new data showing LC3 protein levels in cortex and cerebellum regions of 2-month-old mice using immunohistochemistry (Fig Supp 3)

3. P62 protein level was significantly elevated shown in both cortex and hippocampus of 2-month-old Hexa-/-Neu3-/- mice compared to WT. It is necessary to analyze the P62 in the cortex and hippocampus using immunohistochemistry.

We added new data showing p62 protein levels in cortex and cerebellum regions of 2-month-old mice using immunohistochemistry (Fig Supp 4)

---

## [Decision Letter · Decision Letter 1]

19 Oct 2022

PONE-D-22-09806R1AUTOPHAGIC FLUX IS IMPAIRED IN BRAIN TISSUE OF TAY-SACHS DISEASE MOUSE MODELPLOS ONE

Dear Dr. Seyrantepe,

Thank you for submitting your manuscript to PLOS ONE. Before your manuscript could be further evaluated, please submit the response to the issues raised by the Academic Editor in the previous decision letter.

We look forward to receiving your revised manuscript.

Kind regards,

Vladimir Trajkovic

Academic Editor

PLOS ONE

Reviewers' comments:

Reviewer's Responses to Questions

**Comments to the Author**

1. If the authors have adequately addressed your comments raised in a previous round of review and you feel that this manuscript is now acceptable for publication, you may indicate that here to bypass the “Comments to the Author” section, enter your conflict of interest statement in the “Confidential to Editor” section, and submit your "Accept" recommendation.

Reviewer #2: All comments have been addressed

2. Is the manuscript technically sound, and do the data support the conclusions?

Reviewer #2: Yes

3. Has the statistical analysis been performed appropriately and rigorously? 

Reviewer #2: Yes

4. Have the authors made all data underlying the findings in their manuscript fully available?

Reviewer #2: Yes

5. Is the manuscript presented in an intelligible fashion and written in standard English?

Reviewer #2: Yes

6. Review Comments to the Author

Reviewer #2: The authors have addressed all my concerns. I suggest this manuscript is acceptable for publication.

7. PLOS authors have the option to publish the peer review history of their article (what does this mean?). If published, this will include your full peer review and any attached files.

Reviewer #2: **Yes: **Zhanjun Li

---

## [Author Response · Author response to Decision Letter 1]

6 Dec 2022

We did not see the comments from the reviewer 1.

---

## [Editor Report · Decision Letter 2]

20 Dec 2022

PONE-D-22-09806R2AUTOPHAGIC FLUX IS IMPAIRED IN BRAIN TISSUE OF TAY-SACHS DISEASE MOUSE MODEL

PLOS ONE

Dear Dr. Seyrantepe,

Thank you for submitting your manuscript to PLOS ONE. Before your manuscript could be further evaluated, please submit the response to the issues raised by the Academic Editor in the first decision letter (please see below):

In some of the LC3 blots (e.g. Fig. 4J) there are three clearly visible bands. The authors should explain which of the two lower bands was considered as LC3-II. If possible, the higher quality LC3 blots should be presented in Fig. 4J. Also, the LC3-II levels should be expressed relative to loading control (actin), not LC3-I, as recommended in the last edition of autophagy guidelines (Autophagy 2021;17:1-382. doi: 10.1080/15548627.2020.1797280).

We look forward to receiving your revised manuscript.

Kind regards,

Vladimir Trajkovic

Academic Editor

PLOS ONE
---

## [Author Response · Author response to Decision Letter 2]

4 Jan 2023

Response to reviewers 

Reviewer 1:

This study aims to investigate alterations in autophagic flux in the new mouse model of Tay-Sachs disease, which should provide information for future mechanistic and therapeutic studies in this field. 

1. However, this study only includes 3 mice each groups, lacking statistical rigor.

All data presented here were replicated in at least three mice with all histochemical staining data quantified by experimenters who were blind to genotypes/treatments. Animals were randomly allocated into control and experimental groups.

2. The implications of alterations of those gene expression are not well-explained. 

The rationale why the authors are looking at a certain gene is not fully explained.

The rationale for why looking at a certain gene expression is explained in the introduction section of the manuscript as below

In mammalian cells, two major mechanisms are carried out for the degradation of intracellular proteins which are the ubiquitin-proteasome system (UPS) and autophagy [6]. In particular, autophagy is involved in lysosome-dependent pathways for damaged organelles, unfolded proteins, and accumulated cellular components to maintain cellular homeostasis [7]. Autophagic flux includes vesicle trafficking and a network in which newly produced autophagosomes (double-membrane vesicles) are fused with lysosomes to degrade autophagic cargo. In this process, autophagore formation, autophagosome completion with the closure of membrane, and autophagosome-lysosome fusion (autolysosome) take place respectively [8]. Each step of autophagic flux is finely regulated by specific protein complexes and the initiation step, the sequestering of autophagic cargo within an isolation membrane (phagophore), is controlled by the Beclin-1. Beclin-1 consists of three distinct structural domains interacting with proteins that regulate autophagy machinery. In particular, Beclin-1, which can interact with several cofactors, promotes the formation of the Beclin-1-Vps34-Vps15 core body, in this way induces the initiation of autophagic flux [9]. Autophagy-related (Atg) proteins generate phagophore assembly sites and enable the envelopment of cytoplasmic material. In particular, the Atg9 protein is a key regulator of autophagy induction. During the maturation of autophagosome, Atg7 has involved the conversion of the cytosolic form of microtubule-associated protein 1 light chain 3 (LC3-I) to LC3-II which is located on both inner and outer autophagosomal membranes [9]. LC3 is involved in phagophore edge folding that resulted in autophagosome formation. After the formation of autolysosomes formed by the combination of autophagosomes and lysosomes, LC3-II on the outer membrane is converted back to LC3-I and then intra-autophagosomal LC3-II is degraded by lysosomal hydrolyses. Therefore, the level of LC3-II as a marker of dynamic autophagosomal membranes is generally studied to monitor autophagic activity [10]. In addition, the level of ubiquitin-binding scaffold protein p62 (aggregated endogenous substrates) which know to be associated with LC3-II in the autophagosome is mostly evaluated as a marker of termination of autophagy [9,11]. A bunch of studies also showed a defect in autophagic flux and secondary accumulation of autophagic substrates such as autophagosomes in several LSDs [6,12–14].

 It is difficult to follow why change of one gene in a certain brain region means something for a better understanding of pathogenesis mechanisms of this mouse model. 

Moreover, how the information collected in this study will contribute to a better understanding of Tay-Sachs disease and this mouse model is not clear.

Basically, changes of several genes in some brain regions (but not others) at a certain time is only an indicator (weak evidence) of something. 

This study lacks a clear logic, strong evidence, and a meaningful conclusion. 

In this study, we show that abnormal GM2 accumulation in the different region of brain tissue from TSD mouse model leads to impaired autophagy evident from attenuated expression of some components of the autophagy network, decreased autophagosome formation, and reduced autophagy flux. We for the first time study the expression of some fundamental autophagic markers (LC3, p62, and Beclin-1) in a TSD murine model by q-RT-PCR, immunohistochemistry and Western blot.

3. The reviewer is not convinced that targeting autophagic flux could be potentially therapeutic, which seems to be a conclusion of this study.

We are sorry for not convincing the reviewer that targeting autophagic flux could be potentially therapeutic in this manuscript. However, we have some preliminary in vitro data indicating lithium as a therapeutic agent could be used to restore dysregulated autophagy in Tay-Sachs cells. That data is not completed yet to add to the manuscript. 

Although additional studies are required to fully characterize the autophagy machinery in the recently generated TSD murine model, this paper suggests impaired autophagy as a new aspect involved in the molecular pathogenesis of TSD. Despite important advances made in the last years in TSD research, the principally studied therapies such as gene and anti-inflammatory therapy and their combination can increase the life expectancy of the treated TSD mice (unpublished data yet) autophagy modulation could be a promising option to be tested in combination therapies to help to achieve a complete TSD phenotype rescue.

4. There are many grammar mistakes.

In this final version of the manuscript, we used a free online writing assistant called Grammarly 

Reviewer 2:

The authors examined autophagic flux initiation, autophagosome number in the elongation, and p62/SQSTM1 in termination in 2-month and 5-month TSD mice model cortex, cerebellum, thalamus and hippocampus by qRT-PCR, Western Blot, and immunohistochemical techniques. 

The data revealed accumulated autophagosomes, indicating impairment in autophagic flux in the brain. 

This work elucidates the contribution of altered autophagy to TSD pathology. I would like to see the following points addressed in the manuscript before its publication.

1. Hexa-/-Neu3-/- undergo progressive neurodegeneration and survived up to 5 months. However, no significant difference in the protein level of Beclin-1 was detected in 2-month-old WT, Hexa-/-, Neu3-/-, and Hexa-/-Neu3-/- mice. Thus, more time point data is necessary to determine when autophagic flux alteration in those gene KO mice.

We added one more time point (3-months old mice) to determine when autophagic flux alterations (Fig Supp 1 and Fig Supp 2) 

2. LC3 protein level in cortex and cerebellum regions of 2-month-old Hexa-/-Neu3-/- mice was notably elevated compared to WT. It is necessary to analyze the number of LC3(+) vesicles in cortex and cerebellum using immunohistochemistry.

We added new data showing LC3 protein levels in cortex and cerebellum regions of 2-month-old mice using immunohistochemistry (Fig Supp 3)

3. P62 protein level was significantly elevated shown in both cortex and hippocampus of 2-month-old Hexa-/-Neu3-/- mice compared to WT. It is necessary to analyze the P62 in the cortex and hippocampus using immunohistochemistry.

We added new data showing p62 protein levels in cortex and cerebellum regions of 2-month-old mice using immunohistochemistry (Fig Supp 4)

---

## [Editor Report · Decision Letter 3]

5 Jan 2023

AUTOPHAGIC FLUX IS IMPAIRED IN BRAIN TISSUE OF TAY-SACHS DISEASE MOUSE MODEL

PONE-D-22-09806R3

Dear Dr. Seyrantepe,

We’re pleased to inform you that your manuscript has been judged scientifically suitable for publication and will be formally accepted for publication once it meets all outstanding technical requirements.

Kind regards,

Vladimir Trajkovic

Academic Editor

PLOS ONE
---

## [Editor Report · Acceptance letter]

7 Mar 2023

PONE-D-22-09806R3 

AUTOPHAGIC FLUX IS IMPAIRED IN THE BRAIN TISSUE OF TAY-SACHS DISEASE MOUSE MODEL 

Dear Dr. Seyrantepe:

I'm pleased to inform you that your manuscript has been deemed suitable for publication in PLOS ONE. Congratulations! Your manuscript is now with our production department. 

Kind regards, 

on behalf of

Prof. Vladimir Trajkovic 

Academic Editor

PLOS ONE